# Three-Dimensional Measurement of the Uterus on Magnetic Resonance Images: Development and Performance Analysis of an Automated Deep-Learning Tool

**DOI:** 10.3390/diagnostics13162662

**Published:** 2023-08-12

**Authors:** Daphné Mulliez, Edouard Poncelet, Laurie Ferret, Christine Hoeffel, Blandine Hamet, Lan Anh Dang, Nicolas Laurent, Guillaume Ramette

**Affiliations:** 1Service d’Imagerie de la Femme, Centre Hospitalier de Valenciennes, 59300 Valenciennes, France; poncelet.edouard@gmail.com (E.P.); blandine.hamet@gmail.com (B.H.); dng.lananh@gmail.com (L.A.D.); laurent-n@ch-valenciennes.fr (N.L.); ramette.g@gmail.com (G.R.); 2Unité de Recherche Clinique, Centre Hospitalier de Valenciennes, 59300 Valenciennes, France; ferret-l@ch-valenciennes.fr; 3Service de Radiologie, Hôpital Maison Blanche, Avenue du Général Koenig, 51092 Reims, France; choeffel-fornes@chu-reims.fr

**Keywords:** deep learning, convolutional neural network, artificial intelligence, uterus, measurement, MRI

## Abstract

Uterus measurements are useful for assessing both the treatment and follow-ups of gynaecological patients. The aim of our study was to develop a deep learning (DL) tool for fully automated measurement of the three-dimensional size of the uterus on magnetic resonance imaging (MRI). In this single-centre retrospective study, 900 cases were included to train, validate, and test a VGG-16/VGG-11 convolutional neural network (CNN). The ground truth was manual measurement. The performance of the model was evaluated using the objective key point similarity (OKS), the mean difference in millimetres, and coefficient of determination R^2^. The OKS of our model was 0.92 (validation) and 0.96 (test). The average deviation and R^2^ coefficient between the AI measurements and the manual ones were, respectively, 3.9 mm and 0.93 for two-point length, 3.7 mm and 0.94 for three-point length, 2.6 mm and 0.93 for width, 4.2 mm and 0.75 for thickness. The inter-radiologist variability was 1.4 mm. A three-dimensional automated measurement was obtained in 1.6 s. In conclusion, our model was able to locate the uterus on MRIs and place measurement points on it to obtain its three-dimensional measurement with a very good correlation compared to manual measurements.

## 1. Introduction

Several variations can be observed in the size of the female genitalia, especially the uterus [1]. First, there are individual physiological factors such as age (pre-pubertal phase, time of genital activity, menopause) and such as the natural changes in the aspect of the uterus occurring during gestation. A uterine deformation could also be caused by personal endometrial or myometrial pathologies. Ultimately, there is a significant inter-individual variability depending on everyone regarding the size of the uterus [2].

Uterus measurements are undoubtedly useful for assessing both the treatment and follow-ups of gynaecological patients. Evaluation of the size of the uterus helps describe the development and senescence of the organs, choose the best procedures, and assist surgical procedures such as laparoscopy or laparotomy.

From a clinical perspective, knowing the size of the uterus is useful to diagnose delayed or precocious puberty [3]. Data on the sagittal uterine length and body-to-cervix ratio can be used to conclude whether the internal genitalia are of pubertal or non-pubertal morphology [4]. Also, describing the size of the uterus in the case of polymyomatous pathology is an indicator of the position of the uterus in the patient’s abdomen and enables clinicians to appreciate the potential urinary and digestive repercussions [5]. Furthermore, measuring total uterine length is essential before inserting an Intra Uterine Device (IUD). It helps to determine where to stop IUD insertion and avoid insertion problems such as perforation [6]. The width of the uterine cavity is measured to ensure that T-shaped devices fit properly to the cavity, thus avoiding dimensional disproportions that could lead to problems of effectiveness or material displacement [7]. 

From a surgical perspective, knowing the size of the uterus helps to estimate the best approach for removing the organ when a hysterectomy may be indicated (laparoscopy, laparotomy, vaginal approach...) [8]. Also, it is necessary to measure the transverse diameter of the uterus before endometrial ablation by thermal balloon, cryotherapy, radiofrequency or microwave energy to avoid damaging the endocervical canal [9]. For the NovaSure electrosurgical technique in particular, the radiofrequency controller needs to know the values for the intercornal width of the uterine fundus and the length of the uterine cavity, which must be between 40 and 65 mm [1]. Techniques for treating genital prolapse by promontofixation will also depend on the size of the uterus [10]. 

In the obstetrical field, measuring uterine length provides important information to monitor the progress of the pregnancy and to know whether fetal growth is progressing correctly [11]. If the size of the uterus is too small or too large, this may lead to an ultrasound check on fetal growth, the amount of amniotic fluid, and the appearance of the placenta. Measurements of the uterus can help decide whether curettage is necessary in the event of a failed pregnancy (spontaneous miscarriage or voluntary termination of pregnancy). As the width of the endometrial cavity is correlated with gestational age, measuring it at the time of abortion could help avoid many surgical procedures [12]. Researchers have postulated that by using the mean of cavity width and area plus two standard deviations as the upper limit of cavity size (width 50 mm, area 60 cm^2^), then only 44% of patients in their study would have required curettage [1]. Measuring uterine length is also particularly useful for locating the most suitable intrauterine site for the development of fertilized eggs for in vitro fertilisation. A study by Egbase et al. showed that implantation and pregnancy rates were higher when the total length of the uterus was between 70 and 90 mm [13]. They placed the embryos 5 mm from the uterine fundus.

For all these reasons, an accurate description of uterine measurements must be made in the magnetic resonance (MR) examination of the female pelvis [14].

Usually, the measurement is performed manually by radiologists, wasting medical time and leading to inter-operator variability [15].

Artificial intelligence (AI) is a scientific discipline based on the premise that the faculties of human intelligence can be simulated by a machine. Machine learning consists in creating algorithms capable of learning how to model functions and make predictions using a process of generalisation of the data induced. Deep learning is a sub-type of machine learning, using neural networks that are particularly effective in image processing, as they can learn spatial determinants.

Artificial intelligence has become a potential solution to assist segmentation [16,17]. Thus, AI can develop an automatic segmentation tool for given structures and enable significant improvements in radiological workflows [18]. 

Deep learning models have been successful in providing automatic segmentation for different organs such as the prostate, kidneys, and heart [19,20,21,22].

Measuring the uterus is a more challenging task due to anatomical variability and complex contrasts with surrounding tissues, but also in relation to pathologies such as endometriosis or myomas that distort the contours of the uterus [11]. 

Along with ultrasound, magnetic resonance imaging (MRI) is the best imaging modality for the diagnosis of pelvic pathologies in women. Furthermore, it provides better reproducibility because it is not observer dependent.

A previous study presents a method for automatic uterus segmentation in MRI for patients with uterine fibroids undergoing ultrasound-guided HIFU therapy. They used a 3D nnU-Net model in order to automate uterine volumetry for tracking changes after therapy [23].

Another recent study showed that using a combination of deep learning reconstruction and a sharpening filter markedly increases the image quality of SSFSE of the uterus to the level of the PROPELLER sequence [24].

The aim of our work was to develop, validate, and test an automated deep learning tool for MR images, to implement an automatic measurement of the three-dimensional size of the uterus, and to evaluate its performance compared with the manual measurements of radiologists.

## 2. Material and Methods

This retrospective study of model creation was approved by the Independent Ethics Committee under the reference CHV-2022-006. All patients were informed of the use of their medical data according to the legal framework imposed by CNIL MR-004. In addition, all data were pseudonymized beforehand.

### 2.1. Data Acquisition

All women over 18 years of age who underwent pelvic MRI, including sagittal and axial T2-weighted images, in the women’s imaging department of Valenciennes Hospital (France) between September 2021 and March 2022 were retrospectively collected from the Institutional Picture Archiving and Communication System (Electronic Medical Record Entreprise, VEPRO AG, Version 8.2, Pfungstadt, Germany).

Pregnant women were excluded, as were MR images with severe motion artefacts, a highly deviated uterus, and subserous myomatous pathology (FIGO VI and VII, due to important deformation of the uterus). 

These examinations were performed using two MR units, either at 1.5 T (SIGNA artist) or 3 T (SIGNA premier) (General Electric Healthcare, Cleveland, OH, USA). The acquisition parameters are listed in Table 1.

Patients were randomly assigned to training (80%) and validation sets (20%) without any overlap.

An additional set of MR images acquired between July and August 2021 was used for external validation using the same inclusion and exclusion criteria (test set).

### 2.2. Data Labelling

One radiologist (DM) used artificial intelligence software specifically designed for medical imaging (Cleverdoc V1.9.0 platform, Cleverdoc Entreprise, Lille, France) for labelling. 

When the uterus was not seen, a label of “not observable” was attributed to the examen. When visible, it was targeted by a square area (a label box was drawn on the uterus). Three consecutive cuts were annotated with points corresponding to the measurements of the uterus (Figure 1). Thickness and length were noted on sagittal T2-weighted images. The length was measured using two different methods: one major axis defined by two points, and the other defined by three points passing through the fundus, the cervix-isthmus junction, and the exocervix (Figure 2). Radiologists often prefer the three-point measurement when the uterus is significantly flexed. The width was labelled on the axial T2-weighted images. These manual measurements were used to define the ground truth.

### 2.3. Model

The overall pipeline works due to two separate models working successively. 

First, a box model finds the uteri. Then, a key point model is used to place the measurement points on the cropped version of the image by placing two points for each class. This is achieved by splitting each class into two subclasses. The point for each subclass is determined by its position (top-left or bottom-right). For a length composed of three points, the midpoint is given in a separate preprocessing step. 

We used convolutional neural network (CNN) architectures with an encoder or decoder pattern. 

Encoder

The encoder network was composed of VGG-16 (box model) and VGG-11 (key point model). It receives an image with a size of 224 × 224 pixels as input. Subsequently, it passes through five blocks which are separated by a Max Pooling layer (Kernel = 2.2; Stride size = 2; No padding) that halves the height and width of the features (Figure 3). 

Decoder 

Our model’s decoders are single-instance boxes and key point detectors, which produce one box and one key point instance for each output class, respectively. 

The input of the head is a four-dimensional vector of shape. The outputs are four values per class (x, y, w, h) for the position of the box and two values per class (x, y) for key point positions. 

### 2.4. Training

We applied data augmentation techniques such as vertical and horizontal flipping, random rotation of a multiple of 90°, and translation of up to 0.1. Furthermore, for the key point model, we applied more techniques, such as changing the brightness and contrast, blurring the image, applying Gaussian noise, or reversing the image’s colours. 

We sorted our data based on the classes we wanted to train in (“uterus” classes for the box model and “keypoint” classes for the key point model). We applied a balancer that always kept a batch size of 10 with an equivalent number of items for each class in order to ensure the proper distribution of losses and metrics.

The box model was run for 100 epochs (220,783 iterations), and the key point model was run for 250 epochs (40,320 iterations).

The Adam optimiser was used with a learning rate of 0.0001 to optimise the weight of the model.

### 2.5. Statistics

#### 2.5.1. Model Training

We kept track of the model’s losses and calculated the metrics: The intersection over union (IoU) to evaluate the accuracy of box positions and objective key point similarity (OKS) for each key point class. This metric quantifies the closeness of the predicted key point location by using the ground-truth key point. The closer the predicted key point is to the ground truth, the closer the OKS approach is to 1. Above 0.80, the model is considered very good. This metric is calculated as follows:(1)$$\mathrm{OKS}=exp(-\frac{d^{2}}{2s^{2}k^{2}})$$
where *d* is the distance between the ground truth key point and predicted key point, *s* is the area of the bounding box divided by the total image area, and *k* is the per-key point constant that controls the fall-off.

#### 2.5.2. Model Testing

Four experts with experience in genital imaging (EP, GR, BH, and LD with 12, 2, and 1 years of experience, respectively) manually measured the size of the uterus in three dimensions on every MR image of the test set. The radiologists were blinded to the measurements made by others and to the machine. The same viewer (ViewerCleverdoc1.9.0) was used.

To evaluate the performance of our deep learning tool, we calculated the absolute average difference (in millimetres) between the measurements of one dataset and those of another. The coefficient of determination R^2^ was used to reflect how well the AI measurements matched those of the radiologists using a linear regression model. If R^2^ is equal to 1, the algorithm obtains strictly identical measurements to those of the radiologists. 

The Cleverdoc V1.9.0 platform (Lille, France) was used for the statistical analyses. 

## 3. Results

A total of 845 MRI scans were collected. Moreover, 45 patients were excluded: 37 because of myomas, 6 because of pregnancies, 3 because of a highly deviated uterus, and 2 because of poor image quality. The characteristics of the patients are shown in Table 2.

From the 800 included patients for training and validation, 4800 sets were obtained (three consecutive slices centred on the uterus for each sagittal and axial sequence).

An additional external cohort of 100 MR images was used for the model testing (Figure 4).

### 3.1. Validation Performance (Initial Dataset)

During the validation phase, the algorithm was able to locate the uterus and the measurement key points with excellent accuracy.

With the measurement by DM as ground truth, the mean OKS was 0.92, ranging from 0.90 and 0.94 (Table 3). The OKS was calculated using the cropped images obtained using the box model.

### 3.2. Test Performance (External Dataset)

We observed an improvement in the accuracy of our model when we switched to a new unknown cohort for testing. With the average of radiologists’ measurements as ground truth, the mean OKS of our DL tool was 0.96, ranging from 0.95 and 0.98, as reported in Table 4. The OKS was calculated using full-size images.

Regarding the execution speed, it took less than 5 min for the model to extract all measurements, that is, one three-dimensional measurement in approximately 1.6 s. In comparison, the average time for a three-dimensional measurement by a radiologist was clocked at 37.89 s.

#### 3.2.1. Correlation between Manual and Automated Measurements

Out of the 100 MR images of the test set, the average deviation between AI measurements and radiologists’ measurements was 3.6 mm (±6.6 standard deviation (SD)). The distribution of the gaps was as follows: 3.9 mm for two-point length, 3.7 mm for three-point length, 2.6 mm for width, and 4.2 mm for thickness (Figure 5). The details of the absolute measurements are listed in Table 5.

The R^2^ coefficients of determination between the algorithm’s measurements and the average of the radiologists’ measurements were 0.93 for two-point length, 0.94 for three-point length, 0.93 for width, and 0.75 for thickness, as shown in Figure 6. 

#### 3.2.2. Variability between Measurements by Radiologists

The mean difference in measurements between all radiologists was 1.4 mm, detailed as follows: 1.27 mm for two-point length, 2.2 mm for three-point length, 1.14 mm for width, and 0.93 mm for thickness. The differences between each measure by one radiologist and the average of the others are presented in Table 6.

We performed a secondary analysis to highlight the distribution of the AI errors. Nineteen out of 200 images (100 axial + 100 sagittal) had an absolute deviation (averaged over all image measurements) discreetly greater than 8 mm. This represents less than 10% of the total number of examinations.

## 4. Discussion 

We successfully achieved our goal of developing an artificial intelligence algorithm that is able to locate the uterus in pelvic MR examinations, place measurement key points on it, and provide its three-dimensional measurement with satisfactory accuracy. 

The OKS was close to 1, improving from 0.92 (validation) to 0.96 (test). These results can be explained by the fact that the OKS of the validation phase were calculated based on the cropped images, whereas those of the test phase were calculated from the full-size images. The larger the image, the smaller the positioning error. 

However, the algorithm remains highly performant when encountering new brand images. One of the strengths of our study is that our network was tested in an external cohort which did not have a selection bias applied, except for subserous myomas. This performance favours the generalisation of this model.

To the best of our knowledge, only one study of segmentation of images of the uterus has been conducted to date. Kurata et al. evaluated a U-net architecture to contour the uterus on MR images in the sagittal plane [25]. They reached an average DSC score (dice similarity coefficient, which can be compared to OKS) of 0.82. This study included 122 patients with or without uterine disorders. Our model was optimised by using a substantially larger training database of 800 patients.

In parallel, for men, a wide range of studies have been carried out on the automatic segmentation of images of the prostate, with similar results [26,27]. For example, Alexander Ushinsky et al. trained a customized hybrid 3D-2D U-Net CNN architecture on manually segmented MR images and had a DSC score of 0.898 [28].

However, it is more complex for an AI tool to locate and segment images of the uterus than of the prostate because the uterus can have different positions, bends, or shapes. Moreover, the uterus is surrounded by many elements (colon, bladder, and ovaries). 

Another highlight of our study is that our training dataset was strengthened by the clinical heterogeneity of its cases, both in terms of pathological conditions and patient preparation. It included examinations for cervical cancer, endometriosis, and an ultrasound gel. This suggests that the performance of our CNN would be robust in prospective clinical settings.

Most studies on automated segmentation have used volumetric models or U-Net architectures [29,30]. In contrast, our network’s performance was achieved with the VGG-11/16 architectures. This is a major aspect that led to the success of our algorithm. This model is more suitable for distance measurements because it is specifically designed to locate an organ and place measurement points on it. To do so, our pipeline operates using two different models (box and key point models).

The average deviation between the AI measurements and those of the radiologists was 3.6 mm (±6.6 SD), while the inter-radiologist variability was 1.4 mm. However, the R^2^ coefficient was approaching 0.94 for lengths and width, meaning the coherence remained extremely strong between the radiologists and AI. For thickness, however, the R^2^ coefficient was 0.75, owing to the algorithm being challenged by the junctional zone in rare cases.

The speed of our system is a major advantage over the time required for manual segmentation. In our experience, it takes a radiologist 37.89 s to measure a uterus in three dimensions, set against 1.6 s for the algorithm. Our VGGnet may increase the throughput.

Our algorithm has the ability to overthrow a basic task, thus saving radiologists time for significant intellectual tasks.

Our study had a few limitations that should be acknowledged. First, this was a retrospective, monocentric study. The database was created using three MRI scanners (General Electric Healthcare, Valenciennes Hospital, France). The generalisability to other centres or MRI equipment has not yet been established. We subsequently included images obtained using the same T2-weighted acquisition protocols. We can imagine a comparative study of the performance of the algorithm between different MRI parameters or protocols.

In this first approach to developing a three-dimensional measurement software, we preferred to exclude pregnant women to make easier the algorithm training. In a further study, we could try to include examinations with pregnant uteruses to make our algorithm more inclusive. This could provide useful information for obstetricians.

We can easily imagine a clear application of our AI tool in daily practice. The measurements of the algorithm can be displayed on the image server or automatically added to reports. Valuable information on uterine size could help gynaecologists and surgeons in their daily practice. A patient’s state of genital activity could be precisely described, as well as certain pathological conditions such as myomatous disease. Gynaecological procedures such as IUD insertion, hysterectomy, endometrial resection, or promontoxiation could be facilitated. These uterine measurements could also certainly be used in the obstetrical follow-up of patients.

Subsequent studies are required to prospectively validate our network in a clinical setting. We could consider further studies using the same pipeline to measure endometrial thickness or ovarian dimensions.

## 5. Conclusions 

To conclude, we validated our approach of fully automated measurements of the uterus in MRIs. 

Our VGG-16/11-based convolutional neural network is able to precisely locate the uterus and place measurement key points on it with excellent accuracy. From these points, the three-dimensional measurement of the uterus is obtained. The average difference in measurement between IA and radiologists remains inconsequential, even though it slightly exceeds the inter-radiologist variability.

This provides a useful and performant tool that can easily be applied in clinical practice as an alternative to time-consuming manual tracing. 

## Figures and Tables

**Figure 1 diagnostics-13-02662-f001:**
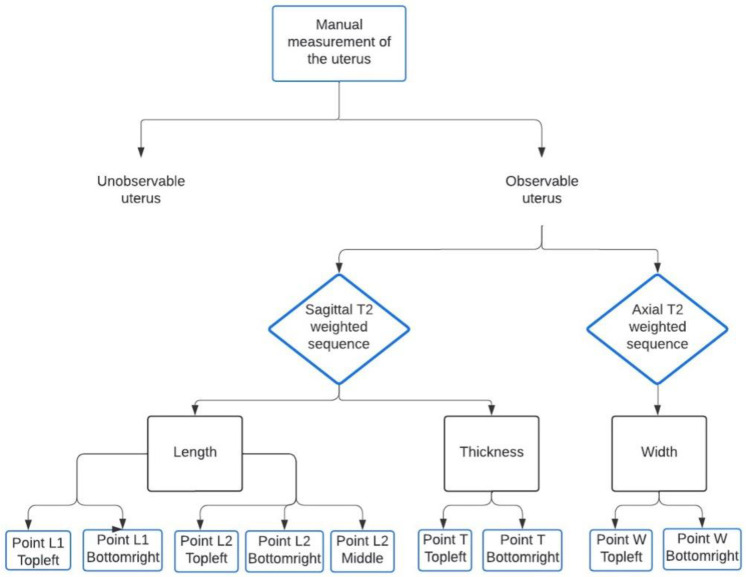
Flowchart of the data labelling.

**Figure 2 diagnostics-13-02662-f002:**
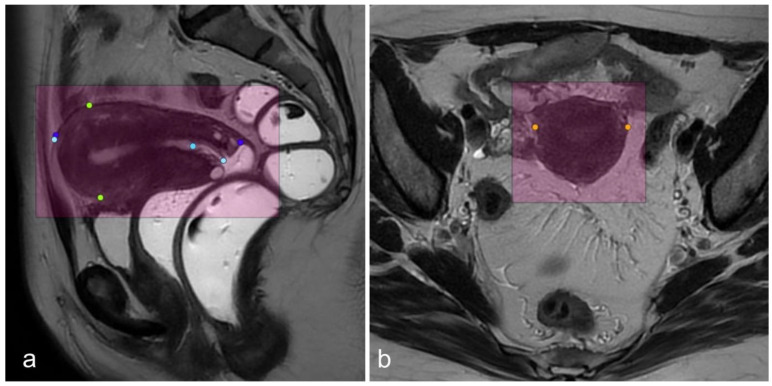
(**a**): Sagittal T2-weighted image with the label “observable” (purple), length label with two points (dark blue), length label with three points (light blue), thickness label (green). (**b**): axial T2-weighted image with label class “observable” (purple) and width label (orange). The vagina and the rectum were filled with ultrasound gel in order to respond to the clinical indication of this case.

**Figure 3 diagnostics-13-02662-f003:**
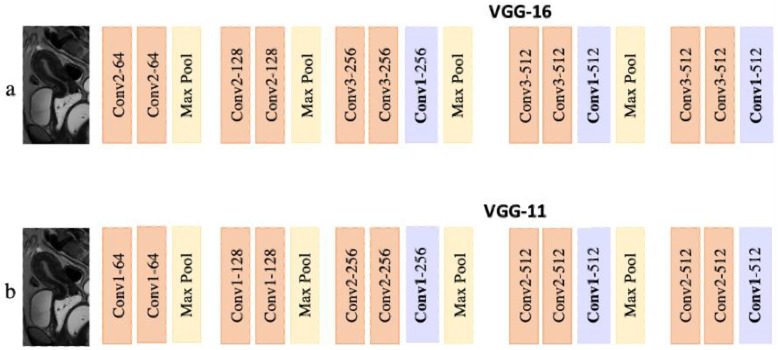
(**a**). VGG-11- modified architecture of the encoder for the box model. (**b**). VGG-16 modified architecture of the encoder for the keypoint model.

**Figure 4 diagnostics-13-02662-f004:**
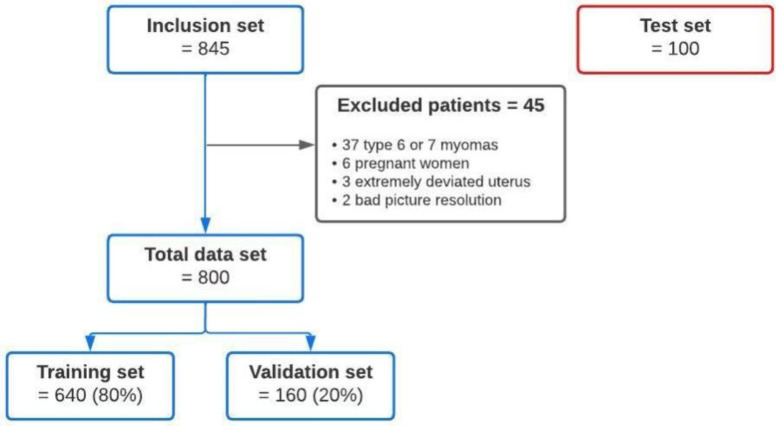
Overview of the workflow for training and testing the automated convolutional neural network (CNN) tool for measurement of the uterus.

**Figure 5 diagnostics-13-02662-f005:**
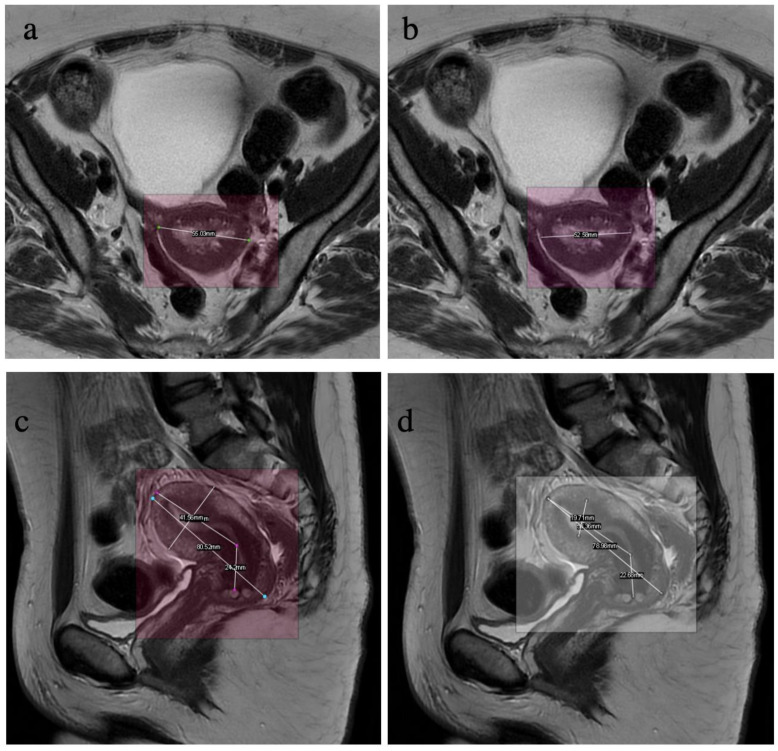
Sample results of the test phase. Width measurements on a T2-weighted axial sequence by radiologist (GR) (**a**) and by the algorithm (**b**). Lengths and thickness measurements on a T2-weighted sagittal sequence by a radiologist (DM) (**c**) and by the algorithm (**d**). We can notice the algorithm’s error in detection of the contours for the thickness measurement.

**Figure 6 diagnostics-13-02662-f006:**
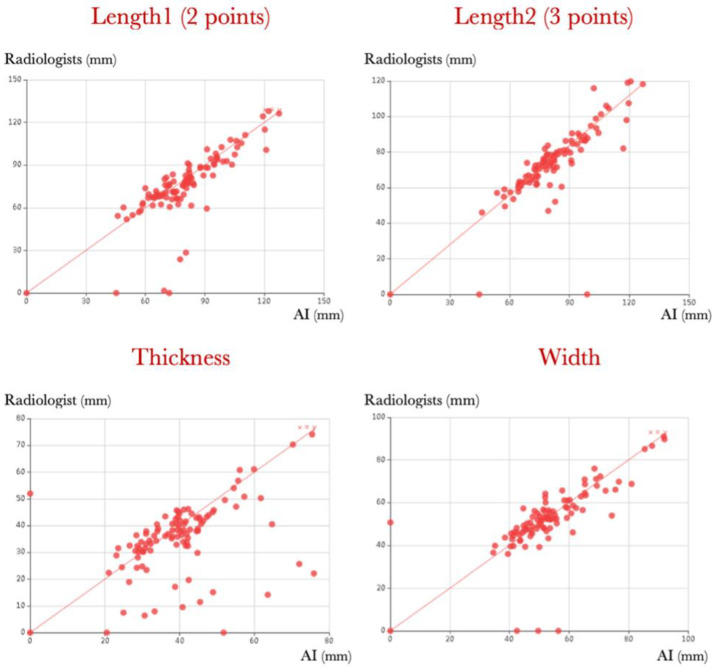
Scatter plots showing the correspondence between artificial intelligence (AI) measurements (in abscissa) and the average of the radiologist’s measurements (in ordinate) in millimetres.

**Table 1 diagnostics-13-02662-t001:** Magnetic resonance imaging (MRI) acquisition parameters.

	MRI	General Electric 1.5 T, SIGNA Artist, 2021		General Electric 1.5 T, SIGNA Artist, 2020		General Electric 3 T, SIGNA Premier, 2019	
Parameters							
Plane		Sagittal	Axial	Sagittal	Axial	Sagittal	Axial
TE (ms)		100–120		100–120		115–120	
TR (ms)		5000–1100		4000–10,000		4000–13,000	
Number of excitations (Nex)		2		2		1.5	2
Field of view (mm)		(393 × 260)– (408 × 220)	363 × 240	393 × 260	(360–240)– (410 × 270)	332 × 220	
Frequency (Hz)		41.67		41.67		50	
Slice thickness (mm)		3.5–4.0		3.5–4.0		3.0–3.5	
Interslice gap (mm)		3.5		3.5		0.5–3.0	

**Table 2 diagnostics-13-02662-t002:** Characteristics of the patients whose data were included for training, validation, and testing of the model.

	Training and Validation Set (*n* = 800)	Test Set (*n* = 100)
Age (mediane (interquartiles))	45 (33–58)	47 (34–56)
Gel vaginal markup		
No	436 (65%)	60 (60%)
Yes	364 (45%)	40 (40%)
Uterus position		
Anteflexed	704 (88%)	93 (93%)
Retroflexed	96 (12%)	7 (7%)
MRI without pelvic pathology	177 (22%)	26 (26%)
Subperitoneal endometriosis	123 (15%)	13 (13%)
Adenomyosis	116 (14%)	12 (12%)
Myomas (FIGO 0—V)	124 (15%)	19 (19%)
Cervical cancer	23 (3%)	2 (2%)
Endometrial pathology	75 (9%)	10 (10%)
Ovarian pathology	165 (21%)	16 (16%)
Hysterectomy	50 (6%)	-
Uterine malformation	7 (0.9%)	1 (1%)
Other (static disorder, no-gynaecological pathology…)	82 (10%)	13 (13%)

**Table 3 diagnostics-13-02662-t003:** Objective key point similarity (OKS) values of the algorithm for each measurement key point, with the measurement by DM as ground truth.

Key Point	Length2 Top left (L1)	Length2 Bottom right (L2)	Length2 Middle (L3)	Length1 Top left (L4)	Length2 Bottom right (L5)	Width Top left (W1)	Width Bottom right (W2)	Thickness Top left (T1)	Thickness Bottom right (T2)	Average (av)
OKS	0.92	0.90	0.94	0.90	0.90	0.94	0.93	0.92	0.93	0.92

**Table 4 diagnostics-13-02662-t004:** Objective key point similarity (OKS) values of the algorithm and of each radiologist, for each measurement key point, with the average of measurements by radiologists as ground truth.

Key point	Length2 Top left (L1)	Length2 Bottom right (L2)	Length2 Middle (L3)	Length1 Top left (L4)	Length1 Bottom right (L5)
GR	0.96	0.96	0.98	0.95	0.95
ED	0.97	0.97	0.98	0.96	0.95
LD	0.96	0.96	0.97	0.97	0.96
BH	096	0.96	0.97	0.96	0.95
AI	0.95	0.95	0.97	0.96	0.95
Key point	Width Top left (W1)	Width Bottom right (W2)	Thickness Top left (T1)	Thickness Bottom right (T2)	Average (av)
GR	0.99	0.98	0.94	0.94	0.96
ED	0.99	0.98	0.95	0.95	0.97
LD	0.99	0.98	0.95	0.95	0.97
BH	0.98	0.97	0.95	0.95	0.96
AI	0.97	0.98	0.95	0.94	0.96

**Table 5 diagnostics-13-02662-t005:** Statistics of uterine dimension measurements by the radiologists (ground truth) and by the algorithm (AI).

	Minimum (mm)		Maximum (mm)		Median (mm)		Average (mm)		Standard Deviation (SD)	
	Ground Truth	AI	Ground Truth	AI	Ground Truth	AI	Ground Truth	AI	Ground Truth	AI
Length 1 (2 points)	44.73	1.59	123.19	127.94	76.46	76.93	79.95	78.09	17.13	19.28
Length 2 (3 points)	42.14	46.04	123.96	119.75	77.23	74.69	79.43	76.19	16.43	15.46
Thickness	19.78	6.3	73.56	74.08	39.23	36.90	39.89	36.23	11.13	11.95
Width	32.16	35.95	93.65	90.98	52.72	52.58	54.42	54.60	12.35	11.06

**Table 6 diagnostics-13-02662-t006:** Average deviation (AD) in millimeters between all radiologists.

Measures	Length2	Length2	Width	Thickness
EP	2.11	1.37	0.97	0.93
LD	2.12	1.2	1.12	0.92
BH	2.23	1.42	1.31	1.08
GR	2.36	1.11	1.15	0.8
Average	2.2	1.27	1.14	0.93

## Data Availability

All data was collected from the Institutional Picture Archiving and Communication System (EMR Manager, VEPRO AG, Version 8.2, Germany) of Valenciennes Hospital, France.

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
