# Peer review of "Three-Dimensional Measurement of the Uterus on Magnetic Resonance Images: Development and Performance Analysis of an Automated Deep-Learning Tool"

_diagnostics, 2023, doi:10.3390/diagnostics13162662_

Round 1

Reviewer 1 Report

In this study, the authors applied the automated deep learning tool to MR images from three-dimensional measurement of the uterus. Although there are several published papers had focused on the deep learning-based evaluation of the placenta and uterus on MR images, this study provides considerable sample size for training and validation. However, there are some major concerns regarding to the study design, inclusion criteria and data interpretation.

1.       Compared to the ultrasound, including 3D ultrasound, measuring the size of the uterus is not the distinctive advantage of 3D MRI. In addition, the measurement of uterus size by deep learning and AI system lacks novelty and advancement in this study. From clinical perspective, MRI can provide more information about uterine cavity volume, deformation, myoma, placental abnormalities, I suggest the authors focus on these fields instead of uterus size measurement.

2.       The representative images and results did not reflect the keynote of three-dimensional measurement.

3.      In Introduction part, the authors listed the necessity of accurate description of uterine measurements in clinical, surgical, obstetrical perspectives, while the authors stated that the cases with pregnancy and severe deformation of the uterus were excluded data acquisition, what was the rationale?  

In this study, the authors applied the automated deep learning tool to MR images from three-dimensional measurement of the uterus. Compared with other published papers focused on the on the deep learning-based evaluation of the placenta and uterus on MR images, including uterine deformation, placental abnormalities, this study provides us with considerable samples, but lacks of novelty only with the size measurement of uterus, I suggest the authors to extend this model for more informative diagnostic fields.  

Author Response

We thank reviewer 1 for his interesting comments.

Point 1. Ultrasound is indeed one of the first-line examinations for measuring the uterus. However, the description of uterine dimensions is also essential in a pelvic MRI report. We focused solely on uterine size and not on malformations. We felt that this three-dimensional approach to uterine size could provide an appreciation of uterine volume.

Point 2. We are sorry if you don't find our results representative of a three-dimensional measurement. The images are extracted directly from the labelling platform and show length, thickness and width measurements.

Point 3. Only Figo 6 and 7 fibromas were excluded from the study. The population included many polymyomatous uteri that were not Figo 6 or 7.
Pregnant women were excluded because we were concerned that, in this first approach to algorithm development, the algorithm would be confounded by the gravid aspect of the uterus. This could be the subject of a future study aimed at improving our algorithm and making it more inclusive. We thank you for this comment. We have added a sentence to this effect in the concluding section of the manuscript (paragraph in red).

The Authors.

Reviewer 2 Report

The current study is a well-written research article on the application of an automated deep-learning tool for Three-dimensional measurement of the uterus using magnetic resonance images. I would like to suggest some minor modifications to improve its quality:

The author should add some sentences about AI and Deep Learning to enrich the introduction section.

Please add previous works in the field using deep learning in the introduction part and if needed compare/contrast in the discussion section for example Deep learning enables automated MRI-based estimation of uterine volume also in patients with uterine fibroids undergoing high-intensity focused ultrasound therapy | Insights into Imaging | Full Text (springeropen.com) and Impact of Deep Learning Reconstruction Combined With a Sharpening...: Ingenta Connect

Author Response

We thank reviewer 2 for his interesting comments.

Point 1. We added some sentences about AI and Deep Learning in the introduction section, as you suggested (paragraphe in red).

Point 2. Thank you for suggesting these two very interesting studies of which we were unaware. We have added the references to these works in the introduction section (paragraph in red).

The Authors.

Reviewer 3 Report

Minor points:
1. The software libraries used for data processig should be indicated and cited (where appropriate)
2. Fixed epoch training for the CNN may not always yield the best results. Early stopping with an appropriate metric monitoring is advised.
3. For the CNN training process, batch size should be indicated.
4. Image downsampling method should be indicated

Author Response

We thank reviewer 3 for his interesting comments.

Point 1. Data was collected from the Institutional Picture Archiving and Communication System (EMR Manager, VEPRO AG, Version 8.2, Germany). The software used for labelling is the Cleverdoc V1.9.0 platform. This information is provided in Sections 2.1 and 2.2.

Point 2. Thank you for this instructive comment. We will apply it in our next studies. 

Point 3. We have added the batch size in Section 2.4.

Point 4. Box model and Keypoint model used both Max Pooling as downsampling method, in order to halve the height and width of the features (Kernel = 2,2 ; Stride size = 2, No Padding). This details have been added in paragraph 2.3. No other downsampling method was applied.

The Authors.